

# Density-dependent condition of juvenile penaeid shrimps in seagrass-dominated aquatic vegetation beds located at different distance from a tidal inlet

Zeferino Blanco-Martínez[1,*], Roberto Pérez-Castañeda[1,*], Jesús Genaro Sánchez-Martínez[1], Flaviano Benavides-González[1], Jaime Luis Rábago-Castro[1], María de la Luz Vázquez-Sauceda[1] and Lorena Garrido-Olvera[2]

[1] Facultad de Medicina Veterinaria y Zootecnia, Universidad Autónoma de Tamaulipas, Victoria, Tamaulipas, Mexico
[2] Instituto de Ecología Aplicada, Universidad Autónoma de Tamaulipas, Victoria, Tamaulipas, Mexico
* These authors contributed equally to this work.

Corresponding author
Roberto Pérez-Castañeda,
roperez@docentes.uat.edu.mx

## ABSTRACT

Seagrasses are critical habitats for the recruitment and growth of juvenile penaeid shrimps within estuaries and coastal lagoons. The location of a seagrass bed within the lagoon can determine the value of a particular bed for shrimp populations. Consequently, differences in the abundance of shrimp can be found in seagrasses depending on their location. As shrimp density increases, density-dependent effects on biological parameters are more likely to occur. However, knowledge about density-dependent processes on shrimp populations in nursery habitats remains limited. The present investigation was undertaken to examine the effects of population density on shrimp condition in two selected seagrass beds, located at different distance from a tidal inlet, one 25 km away (distant) and the other 1 km away (nearby), in a subtropical coastal lagoon. The study was based on monthly samplings during one year in Laguna Madre (Mexico), performing a total of 36 shrimp trawls (100 m$^2$ each one) within each seagrass bed ($n = 3$ trawls per bed per month for 12 months). Shrimp density was related to the proximity to the tidal inlet (higher density was consistently observed in the nearby seagrass bed), which in turn adversely affected the condition of both species studied (*Penaeus aztecus* and *P. duorarum*). In this regard, the present study provides the first evidence of density-dependent effects on shrimp condition inhabiting a nursery habitat. Both shrimp species exhibited a negative relationship between condition and shrimp density. However, this pattern differed depending on the proximity to the tidal inlet, suggesting that shrimp populations inhabiting the nearby seagrass bed are exposed to density-dependent effects on condition; whereas, such effects were not detected in the distant seagrass bed. Shrimp density within the distant seagrass bed was probably below carrying capacity, which is suggested by the better shrimp condition observed in that area of the lagoon. Intra and interspecific competition for food items is surmised to occur, predominantly within seagrass beds near the tidal inlet. However, this hypothesis needs to be tested in future studies.

## INTRODUCTION

Seagrasses are angiosperms adapted to marine and brackish environments that can be found on the bottom of tropical and temperate coastal ecosystems such as bays, estuaries, or coastal lagoons (*Short et al., 2007*). Seagrass can form extensive beds that provide structural complexity to the coastal habitats. Its coverage and density have a positive influence on the abundance and diversity of different taxonomic groups of macrofauna (*McCloskey & Unsworth, 2015*; *Junhui et al., 2018*; *Ruesink et al., 2019*). Seagrasses provide refuge and food for the juvenile phase of many species of fish and crustaceans such as penaeid shrimps (Penaeidae family); accordingly, they are recognized as nursery habitats (*Jackson et al., 2001*).

Submerged aquatic vegetation beds can be constituted by seagrasses, macroalgae, or a mixture of both  (*Haywood, Vance & Loneragan, 1995*; *Githaiga et al., 2016*). Juvenile penaeid shrimps have a marked preference for these habitats, exhibiting generally greater abundance in sites with high density of submerged aquatic vegetation (*Loneragan et al., 1998*; *Pérez-Castañeda et al., 2010*). Additionally, higher levels of aquatic vegetation biomass have been directly related to shrimp growth rates and inversely related to mortality (*Loneragan et al., 2001*; *Pérez-Castañeda & Defeo, 2005*).

Because aquatic vegetation can support a high abundance of shrimp, this could result in competition for resources (e.g., food and shelter) as shrimp density increases, and resources become limited (*Begon, Townsend & Harper, 2006*). Evidence of density-dependence in growth and mortality for shrimp in sheltered coastal habitats indicates that the increase in shrimp abundance leads to decreased growth rates and increased mortality (*Pérez-Castañeda & Defeo, 2005*), which may be due to intra and interspecific competition (*Dahl, Edwards & Patterson, 2019*).

Condition is a biological parameter involving the body weight at a given length; it is related to the availability and consumption of food, reflecting the uptake and allocation of energy (*Lloret, Shulman & Love, 2014*). Shrimp condition can be an indicator of the nutritional status of the organisms; however, it may also vary according to the reproductive cycle, where mature shrimp are heavier than immature ones of the same length (*Chu et al., 1995*). This difference could be due, in part, to the extra weight of the ovaries in mature females, which constitutes up to 13.7% of the total body weight (*Peixoto et al., 2003*). Shrimps found in coastal lagoons and estuaries are primarily sexually immature juveniles (*Dall et al., 1990*); therefore, changes in the condition of juvenile shrimps would mainly be associated to their nutritional status, which can be influenced by habitat characteristics.

Although there are some studies about the condition in juvenile shrimps (*Pérez-Castañeda & Defeo, 2002*; *Ochwada-Doyle et al., 2011*), density-dependence on shrimp condition has been little explored, particularly in nursery areas. While seagrass beds are key habitats for the recruitment of postlarvae and growth of juvenile shrimp, it has been documented that the location of a seagrass bed within the lagoon can determine the value of a particular bed (*Bell, Steffe & Westoby, 1988*). In some coastal lagoons with minimal tidal currents and limited water circulation, like Laguna Madre (Mexico), the distribution of postlarvae within the lagoon could also be limited. In this coastal lagoon, seagrass meadows
nearby a tidal inlet, where the postlarvae enter the lagoon, have a greater abundance of shrimp than distant meadows (*Blanco-Martínez & Pérez-Castañeda, 2017*). Accordingly, it would be expected that these differences in abundance could result in differences in shrimp condition among seagrass beds. However, this issue has not been assessed. This study aimed to evaluate density-dependence in the condition of two juvenile shrimp species (*Penaeus aztecus* Ives, 1891 and *P. duorarum* Burkenroad, 1939) in seagrass-dominated aquatic vegetation beds located at a different distance from a tidal inlet in a coastal lagoon.

## MATERIALS & METHODS

### Sampling and laboratory analysis

This study was conducted in the central part of the Laguna Madre (Mexico), which is a 200 km-long subtropical coastal lagoon located in the Gulf of Mexico (23°50′−25°30′N, 97°15′−97°45′W) (Fig. 1). The hydrography of Laguna Madre is characterized by a microtidal regime (tidal range <0.5 m) exhibiting limited circulation and exchange with waters from the Gulf of Mexico (*Britton & Morton, 1989*; *Tunnel Jr & Judd, 2002*); therefore, minimal tidal currents within the lagoon are generated.

The bottom of the lagoon is characterized by the presence of submerged aquatic vegetation in shallow areas along the coast dominated by submerged seagrasses (*Arellano-Méndez et al., 2019*), which, due to their subtidal nature, are permanently available for aquatic fauna. The last available mapping of seagrasses in Laguna Madre was performed in the 90s using satellite imagery, which displayed seagrass beds distributed at the north-central portion of the lagoon, mostly along the back of the barrier islands (*DUMAC, 1996*). The selected seagrass beds for the present study were located at the central portion of the lagoon. They were situated within the band of seagrasses previously mapped. Similarly, larger seagrass meadows have also been recently reported for the central portion of Laguna Madre (*Arellano-Méndez et al., 2019*); however, there are no data on the distribution of seagrasses at different depth strata in the study area.

The location of sampling sites is shown in Fig. 1. They were situated in seagrass beds located at a different distance (distant and nearby) from a tidal inlet. The distant bed was 25 km from the inlet, while the nearby bed was only 1 km away. The dominant seagrass species in the shrimp nursery habitat were *Halodule wrightii* (77% of total seagrass biomass) and *Syringodium filiforme* (23% of total seagrass biomass) with the presence of some macroalgal species such as *Digenia simplex*, *Penicillus capitatus*, *Jania adherens*, *Laurencia poitei* and *Champia parvula* (*Blanco-Martínez & Pérez-Castañeda, 2017*).

Monthly shrimp samples were collected at night during an annual cycle (January–December 2005) in both seagrass beds (nearby and distant) (Permit from CONAPESCA: DGOPA/05675/060505/.-3869). Data were collected as previously described in *Blanco-Martínez & Pérez-Castañeda (2017)*. Specifically, field collection and laboratory analysis of shrimp and seagrass samples, as well as measurements of salinity, temperature, and dissolved oxygen in the coastal lagoon, were performed as previously described in the study mentioned above. Measurement of carapace length (CL, mm), weighing (body weight, g), and the taxonomic identification of shrimp were also carried out as described therein.

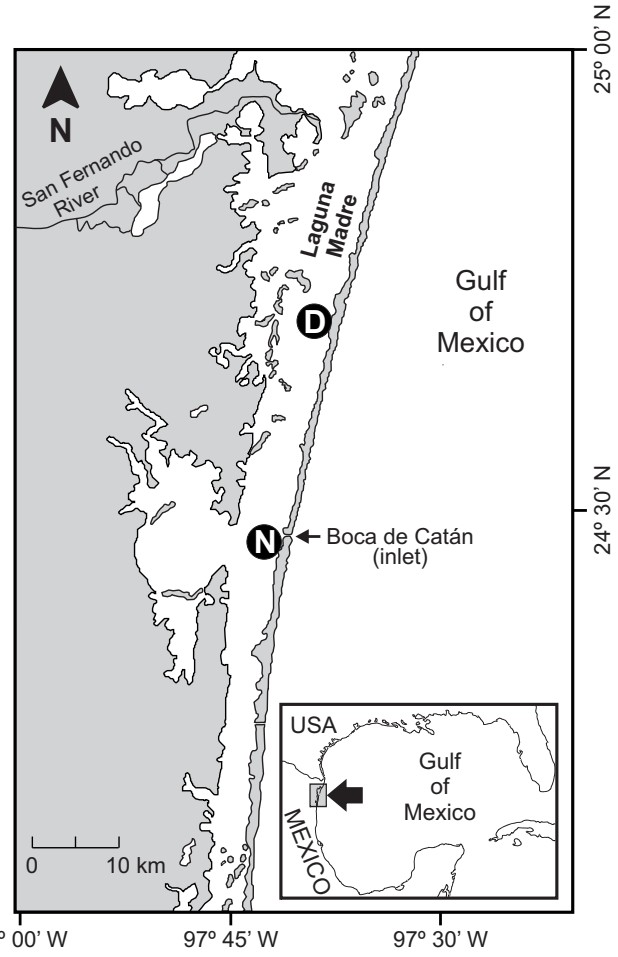

**Figure 1** **Study area.** Location of sampling sites in seagrass beds in Laguna Madre (Mexico). The distant (D) and nearby (N) seagrass beds were located at 25 km and 1 km away, respectively, from the tidal inlet (Boca de Catán).

Individuals under 8.0 mm in carapace length were not identified to species level due to the absence of differentiating morphological characters in those sizes (*Pérez-Farfante, 1970*).

The 2.5 m long beam trawl (1.3 cm mesh size) utilized for shrimp sampling had a 2 m wide and 0.6 m high rigid mouth. The sampling gear was hand-hauled parallel to the water's edge (between 1 and 1.5 m depth approximately), covering a swept area of 100 m$^2$ per tow. Three replicate samples per seagrass bed were taken each month.

## Data analysis

For each seagrass bed, a combined length-frequency distribution (grouped by one mm CL), was constructed for all shrimp sampled in order to depict the overall size structure of shrimp by seagrass bed. The monthly mean abundance of shrimp (total and by species) was also plotted for both sites.

Carapace length-weight (CL-W) relationships for each shrimp species (*P. aztecus* and *P. duorarum*) were determined in each seagrass bed. Weight comparison at a given length

was utilized as a condition indicator under the assumption that heavier individuals of a given length are in better condition, as considered in previous studies (*García-Berthou & Moreno-Amich, 1993*; *Meretsky et al., 2000*). In this regard, an analysis of covariance (ANCOVA) was run on differences in weight at a given length (as an indicator of condition in shrimp) between seagrass beds, with log W as the dependent variable and log CL as the covariate (*Pérez-Castañeda & Defeo, 2002*; *Ochwada-Doyle et al., 2011*). Data fulfilled the linearity assumption between log CL and log W. The homogeneity of slopes (parallelism test) of the fitted linear functions between seagrass beds was also met, allowing the ANCOVA to be carried out. Subsequently, the predicted mean weights for a covariate value of CL = 11.0 mm (corresponding to the average size) were obtained from the ANCOVA to illustrate differences in shrimp condition between seagrass beds for each species.

Additionally, length-weight relationships were obtained by species (*P. aztecus* and *P. duorarum*) fitting the power function $W = a\,CL^b$ for each month and seagrass bed. Afterward, fitted models were utilized to estimate the predicted weight at a given length as an indicator of shrimp condition (in this case weight at CL = 11 mm). Finally, monthly shrimp condition was plotted against the monthly mean abundance of total shrimp, fitting a linear function to identify density-dependence in shrimp condition by seagrass bed. It is worth noting that this analysis was performed with data from the months when shrimp were most abundant, in order to guarantee a better representation of length/weight data sets for fitting the aforementioned power function. In this regard, the months included in the analysis were January–April and November–December. Data for *P. duorarum* from November–December in the distant seagrass bed were excluded because of low abundance.

## RESULTS

Within the lagoon, salinity ranged from 31 to 45, temperature from 16.6 to 30.2 °C, dissolved oxygen from 2.4 to 6.3 mg/L, seagrass biomass from 10.7 to 1,300.7 g/m$^2$, macroalgal biomass from 5.6 to 457.1 g/m$^2$ and the total seagrass biomass from 47.5 to 1,338.3 g/m$^2$. Significantly higher mean salinity (40.6 ± 4.5) and macroalgal biomass (184.6 ± 127.4 g/m$^2$) were registered at the distant seagrass bed. The temperature, dissolved oxygen, seagrass biomass, and the total seagrass biomass were slightly higher at the nearby seagrass bed. However, no significant differences were detected in comparison to the distant bed (Table 1, Supplemental information 1).

Overall, shrimp size ranged from 2.9 to 23.5 mm CL in the distant seagrass bed and from 1.5 to 26.5 mm CL in the nearby seagrass bed. In both vegetation areas, shrimp of 8 to 12 mm CL were the most abundant size classes showing a clear dominance of juveniles (Figs. 2A–2B, Supplemental information 2). On the other hand, the abundance of total shrimp and both species were systematically higher in the seagrass bed nearby to the tidal inlet, both annually and in at least ten months throughout the annual cycle (Figs. 2C–2H, Supplemental information 2).

Shrimp abundance peaked at the beginnin and the end of the year, in the case of total shrimp and *P. aztecus*. However, *P. duorarum* abundance only peaked at the beginning of the year (Fig. 2, Supplemental information 2).

**Table 1  Environmental conditions and submerged aquatic vegetation biomass.** Mean (±SD) values of abiotic factors (salinity, temperature and dissolved oxygen) and for biomass of seagrass and macroalgae, including total biomass, in two seagrass-dominated aquatic vegetation beds located at different distances (distant and nearby) from a tidal inlet in Laguna Madre (Mexico). Probability values of statistical comparisons (Student's $t$ tests) between beds are included.

| | Distant seagrass bed | Nearby seagrass bed | $p$-value |
|---|---|---|---|
| Salinity | $40.6 \pm 4.5$ | $37.4 \pm 2.6$ | 0.042 |
| Temperature (°C) | $24.9 \pm 5.0$ | $25.3 \pm 4.4$ | 0.847 |
| Dissolved oxygen (mg/L) | $4.2 \pm 1.1$ | $4.6 \pm 0.9$ | 0.345 |
| Seagrass (g/m$^2$) | $310.8 \pm 259.1$ | $433.1 \pm 330.6$ | 0.324 |
| Macroalgae (g/m$^2$) | $184.6 \pm 127.4$ | $84.7 \pm 103.4$ | 0.047 |
| Total biomass (g/m$^2$) | $495.4 \pm 319.1$ | $517.8 \pm 311.2$ | 0.863 |

Length-weight relationships, in logarithmic scale, for both shrimp species from each seagrass bed were successfully fitted by a linear function ($\log W = a + b \log CL$) with $r^2$ values > 0.9 in all cases (Fig. 3, Supplemental information 3). Both shrimp species showed homogeneity of slopes (parallelism test) among seagrass beds, as indicated by the interaction term between site and log CL which was not significant ($p = 0.16$ and $p = 0.36$ for *P. aztecus* and *P. duorarum*, respectively; Supplementary information 3). The ANCOVA results indicated significant differences ($p < 0.001$) in weight (log W) at a given length (log CL) between seagrass beds in both shrimp species (Supplemental information 3).

The highest predicted mean weight of shrimps (*P. aztecus* = 0.785 g, *P. duorarum* = 0.789 g) utilized as an indicator of body condition, were observed at the seagrass bed with lowest shrimp density (total abundance = 43.72 ind./100 m$^2$). In contrast, the lowest predicted mean weights (*P. aztecus* = 0.751 g, *P. duorarum* = 0.759 g) were found at the seagrass bed with highest shrimp density (total abundance = 105.61 ind./100 m$^2$; Fig. 4, Supplemental information 3) indicating that shrimp condition decreased as total shrimp abundance increased.

Moreover, when analyzing the data separately by seagrass bed, and merging results from both shrimp species, a negative trend between condition and shrimp abundance was exclusively detected for shrimp inhabiting the aquatic vegetation bed near the tidal inlet (Fig. 5, Supplemental information 4).

Although there were significant differences in salinity and biomass of macroalgae between distant and nearby seagrass beds, these variables were not correlated with shrimp abundance and condition (Supplemental information 4).

## DISCUSSION

Although the size range of shrimp was relatively similar in the distant and nearby seagrass beds, the abundance of shrimp was higher in the seagrass area close to the tidal inlet. This pattern was observed practically all year round, suggesting a possible relationship between shrimp abundance and the proximity to the inlet. The microtidal regime in Laguna Madre generates minimal tidal currents and limited water circulation. This condition might limit the distribution of postlarvae within the lagoon as the distance from the source of postlarvae

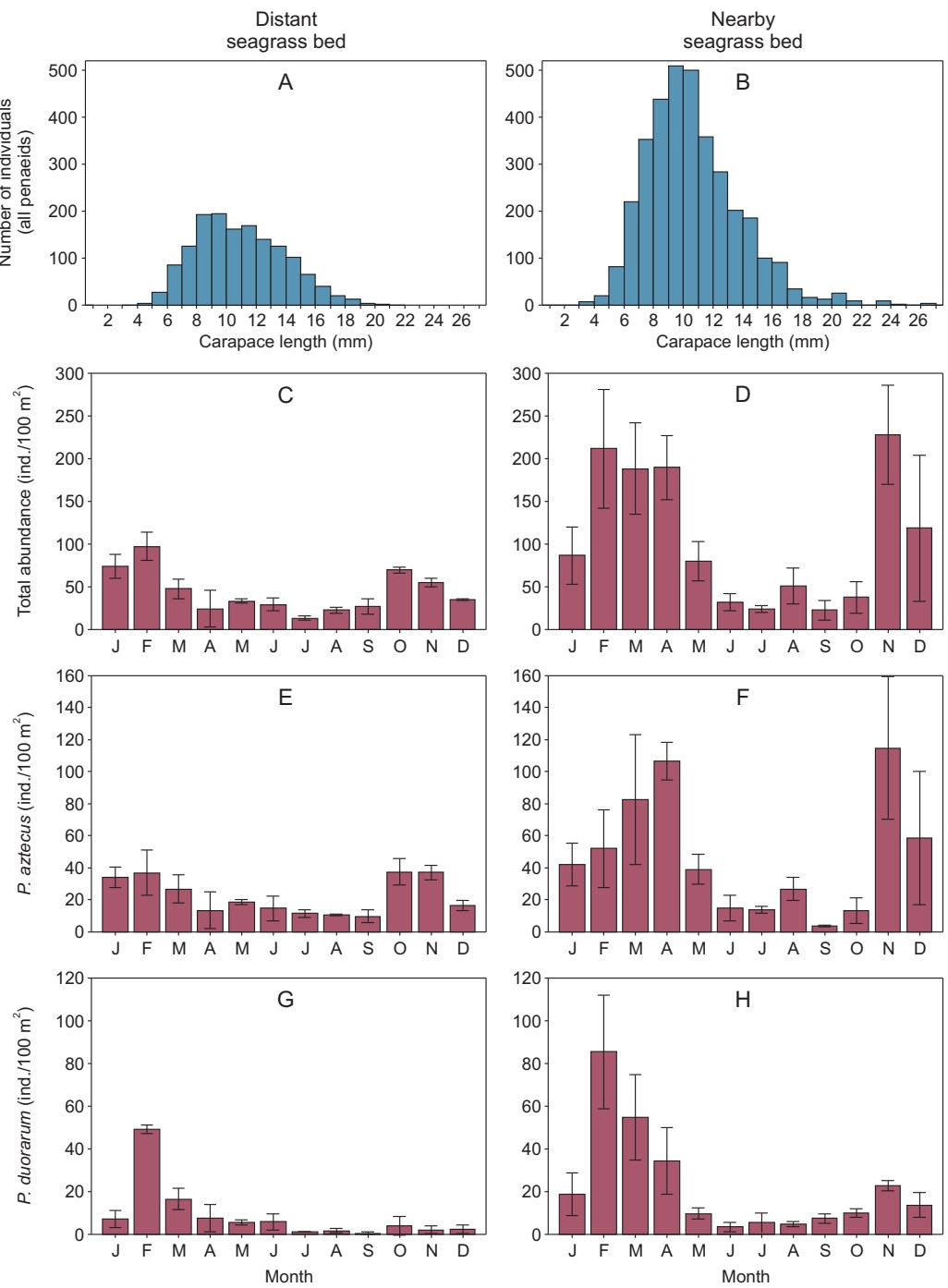

**Figure 2** **Penaeid shrimps collected in two different seagrass beds in relation to their proximity to a tidal inlet (distant and nearby) in Laguna Madre (Mexico).** (A, B) Length-frequency distributions in carapace length (CL) of all penaeids grouped by one mm CL. Monthly abundance (mean ± SD) of shrimp is displayed as (C, D) total abundance, (E, F) *Penaeus aztecus* and (G, H) *P. duorarum*, respectively.

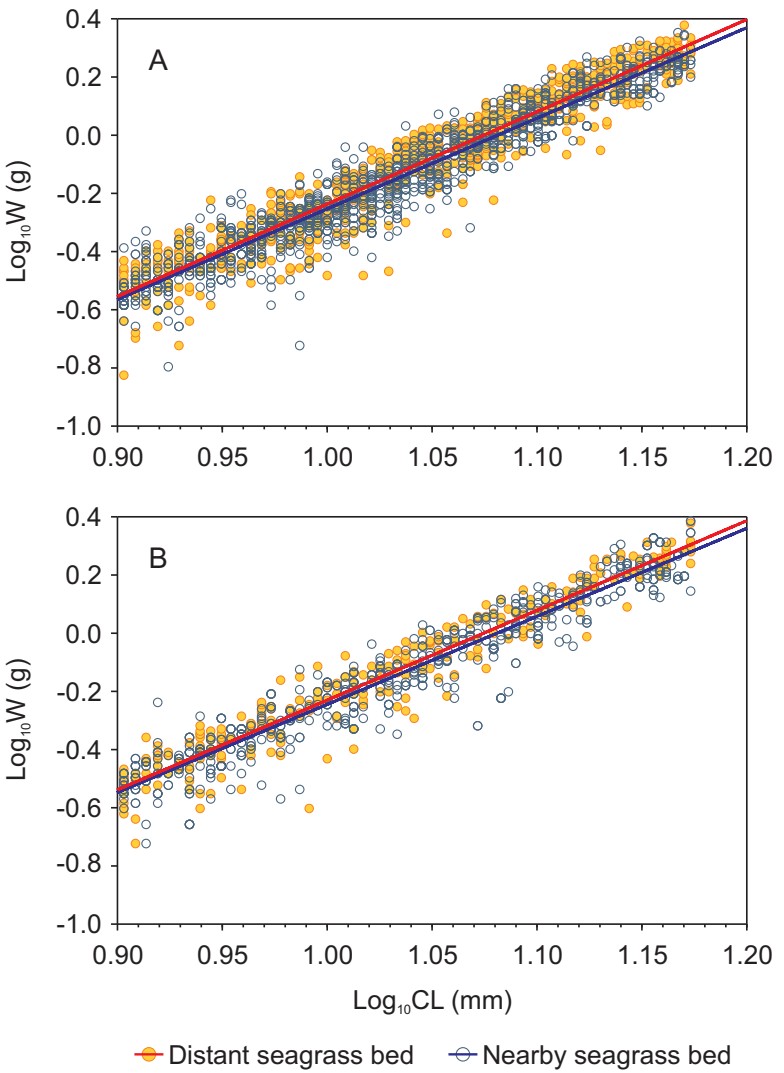

Distant seagrass bed    Nearby seagrass bed

**Figure 3  Carapace length-weight relationships (in logarithmic scale) for juvenile *Penaeus* shrimps from two different seagrass beds, distant and nearby to a tidal inlet, in Laguna Madre.** (A) *P. aztecus*: $r^2 = 0.92$, $n = 685$ in the distant bed, and $r^2 = 0.93$, $n = 1,426$ in the nearby bed; (B) *P. duorarum*: $r^2 = 0.95$, $n = 258$ in the distant bed, and $r^2 = 0.93$, $n = 686$ in the nearby bed.

(tidal inlet) increases. This pattern is consistent with what *Bell, Steffe & Westoby (1988)* pointed out for recently settled decapods of ocean-spawned species; they hypothesized that the abundance of such species is influenced by their location within the estuary. It has been surmised that the proximity to the tidal inlet could influence the value of seagrass habitats for shrimp recruitment and abundance within the lagoon (*Blanco-Martínez & Pérez-Castañeda, 2017*); however, further studies should be carried out to validate this hypothesis.

Because no recently settled postlarvae or very small juvenile shrimp as indicators of recruitment were sampled, no differences in shrimp recruitment was determined between the two seagrass beds. Our study was focused on mid-aged resident juveniles in both

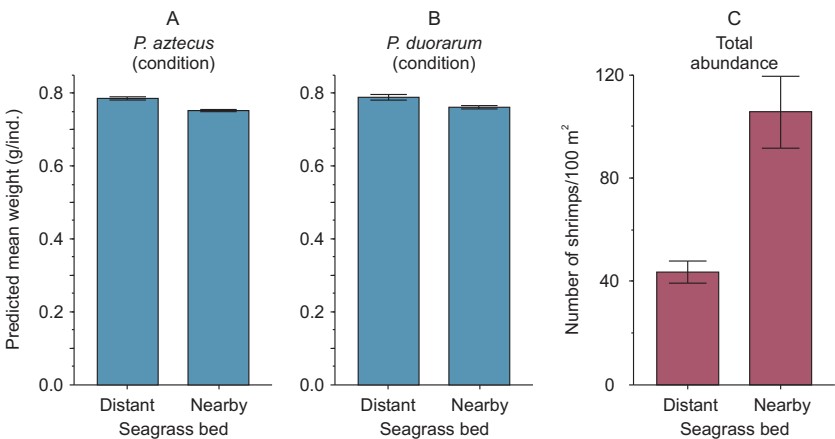

**Figure 4** **Condition and abundance of *Penaeus* shrimps from two different seagrass beds in relation to their proximity to a tidal inlet (distant and nearby) in a coastal lagoon.** (A) Individual predicted mean weight (±SE) at a length of 11 mm CL as an indicator of condition for *P. aztecus* and (B) *P. duorarum*. (C) Mean (±SE) total shrimp abundance.

subtidal seagrass beds. According to the life cycle of penaeids, shrimp in our study (mostly 6 to 15 mm CL; Figs. 2A–2B) were recruited as postlarvae, approximately at 1–2 mm CL. Afterward, they were probably exposed to predation and competition for several weeks and possible redistribution to microhabitats within the seagrass bed before being sampled as part of this study.

The brown shrimp, *P. aztecus*, has also been reported in intertidal salt marsh vegetation from another subtropical coastal habitat (size range: 10–83 mm TL or 2–19 mm CL approximately) (*Minello, Zimmerman & Martinez, 1989*); however, in Laguna Madre salt marsh vegetation is scarce, and therefore, seagrass is the primary habitat with vegetation for penaeid shrimps in this lagoon.

The present study provides the first evidence of density-dependent effects on shrimp condition from different seagrass beds vis-à-vis their proximity to a tidal inlet (the site for the entry of postlarvae into the lagoon). The proximity to the tidal inlet was related to the density of shrimp (higher density was consistently observed in the nearby seagrass bed), which in turn, was negatively related to the body condition.

Shrimp species (*P. aztecus* and *P. duorarum*) in Laguna Madre were subject to density-dependent effects on body condition, as indicted by the negative relationship between predicted mean weight and shrimp abundance, coinciding with that reported for *P. duorarum* in another coastal lagoon (*Pérez-Castañeda & Defeo, 2002*); however, this pattern differed according to the proximity to the tidal inlet, since such negative influence of density on shrimp condition was detected in the nearby seagrass bed but not in the distant bed (Fig. 5). This finding suggests, that shrimp populations inhabiting the nearby seagrass bed are exposed to density-dependent effects on the condition, while on the contrary, such effects are not present in shrimp populations from the distant bed. The determining factor for the detection of density-dependent effects on the condition between the two seagrass areas was the contrasting difference in shrimp abundance.
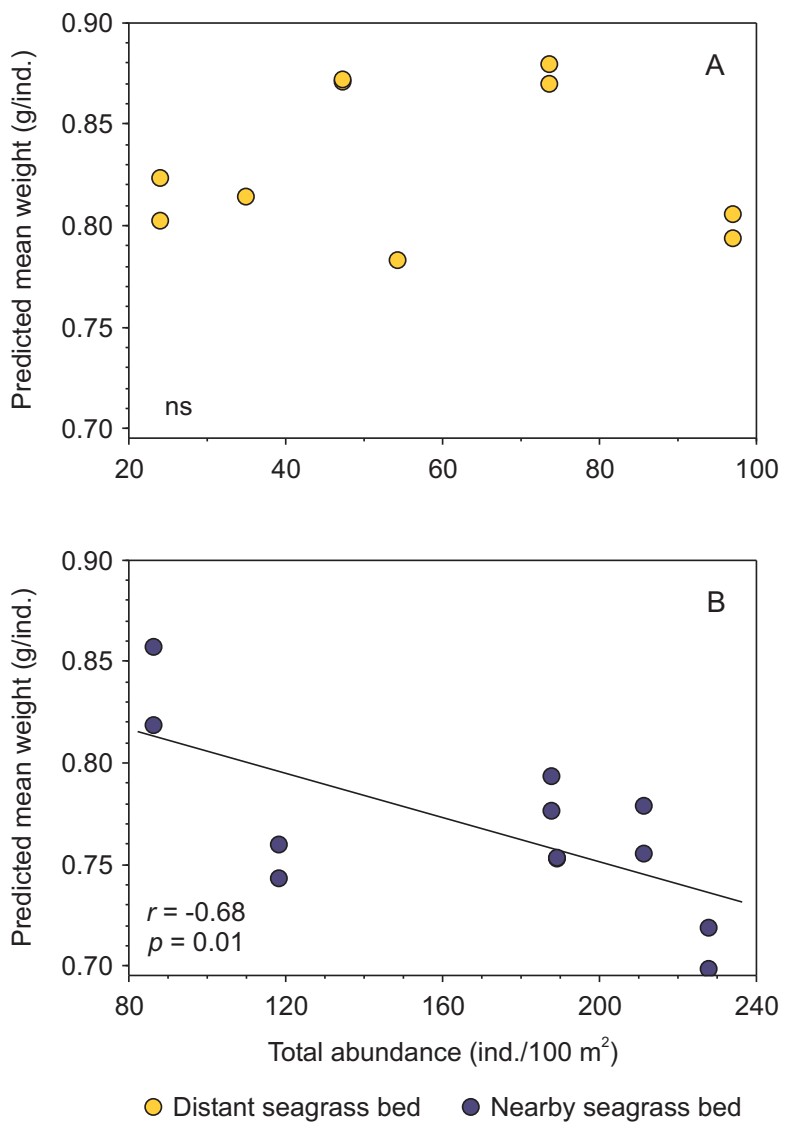

**Figure 5** **Relationship between body condition (combined data from *Penaeus aztecus* and *P. duorarum*) and total shrimp abundance in two seagrass beds, one distant and one near the tidal inlet in a coastal lagoon.** (A) No significant (ns) relationship was detected in the distant bed, whereas (B) a negative linear function between both variables was observed for shrimp in the nearby seagrass bed. Individual predicted mean weight at length of 11 mm CL was utilized as an indicator of condition.

As the condition was density-dependent, growth could also be negatively affected by shrimp density, particularly in the nearby seagrass bed. Although density-dependent growth has been demonstrated for juvenile shrimp in seagrass habitats through field samplings (*Pérez-Castañeda & Defeo, 2005*) and experiments (*Loneragan et al., 2001*), it has not been assessed whether such density-dependent relationship could be related to distance to the source of postlarvae. Unfortunately, in the present study, it was not possible to carry out the identification and tracking of cohorts over time to estimate shrimp growth, as performed by other authors, because more frequent sampling would have been necessary due to the

fast growth of juvenile shrimps before migrating to the sea (*Haywood & Staples, 1993*; *O'Brien, 1994*). Therefore, future studies on the possible effect of density on shrimp growth at different distances from the tidal inlet would be necessary.

Density-dependent effects may be the result of high depletion rates of food sources caused by increments in shrimp density resulting in intra and interspecific competition, as observed in other marine invertebrates (*Gaymer, Himmelman & Johnson, 2002*). In this regard, the two closely related shrimp species (*P. aztecus* and *P. duorarum*) analyzed in the study area (Laguna Madre) represent congeneric and sympatric species that co-occur in seagrass beds, exploiting the same resources (space and food). Thus, both penaeid species may be exposed to intra and interspecific competition, especially as shrimp density increases within the seagrass habitat.

The body condition of postlarval and juvenile shrimp may be indicative of their health status. This biological parameter may even affect their ability to escape from predators, as suggested for *Penaeus plebejus* in Australian coastal lakes, probably resulting in higher mortality for shrimp with a lower condition, i.e., poorer health (*Ochwada-Doyle et al., 2011*).

According to the above, a better body condition of shrimp in the distant seagrass bed in Laguna Madre could imply an advantage for the survival and persistence of shrimp populations within that bed and its possible contribution to the offshore adult's replenishment population. However, the possible effects of body condition on predation should be evaluated.

Although from the site representative point of view, our research was limited (only two sampling beds), it was representative in time (three samples per bed per month for 12 months), observing that differences in shrimp condition between seagrass beds were consistent over the study. Higher data dispersion in the nearby bed could be due to greater shrimp patchiness during the months with peak abundance. However, it would be necessary to evaluate possible differences in shrimp patchiness between seagrass beds in the future, based on a higher number of trawls per bed to differentiate between dispersion due to possible patchiness or low representativeness of the sample size.

Shrimp size classes represented in our study inhabiting the seagrass habitats are representative of the interrelation of growth, mortality, and migration, and whose future survival will likely contribute to the offshore adult population. Although shrimp from the seagrass bed near the tidal inlet had a lower condition, exhibiting density-dependent effects, they were much more abundant and located closer to the adult habitat; i.e., the marine habitat. Therefore, the differential contribution of both seagrass beds (distant and nearby to the tidal inlet) to the adults population should be determined evaluating possible differences in shrimp population dynamics between both beds, including the migration of juveniles to adult habitats (*Beck et al., 2001*). However, such factors have not yet been quantified.

## CONCLUSIONS

Density-dependent effects on body condition were detected in juvenile penaeid shrimps inhabiting seagrass-dominated aquatic vegetation beds. Data indicated that the proximity

to the tidal inlet (site where the postlarvae enter the lagoon) was positively related with the abundance of shrimp, which in turn, was negatively correlated with body condition. This fact was evidenced in both penaeid species (*P. aztecus* and *P. duorarum*). In this regard, intra and interspecific competition by food items is hypothesized to occur, predominantly within the seagrass bed near the tidal inlet. However, this hypothesis needs to be tested in future studies.

## ACKNOWLEDGEMENTS

We thank Mr. Emeterio Dueñez Resendiz, from Carboneras village, for his help during biological sampling in Laguna Madre. This work is part of the doctoral thesis of the first author (ZBM) at the Instituto de Ecología Aplicada, Universidad Autónoma de Tamaulipas.

### Funding

This study was funded by PROMEP Mexico through grants 103.5/04/1405 and 103.5/05/3156 awarded to Roberto Pérez-Castañeda. The funders had no role in study design, data collection and analysis, decision to publish, or preparation of the manuscript.

### Grant Disclosures

The following grant information was disclosed by the authors:
PROMEP Mexico: 103.5/04/1405, 103.5/05/3156.

### Competing Interests

The authors declare there are no competing interests.

### Author Contributions

- Zeferino Blanco-Martínez and Roberto Pérez-Castañeda conceived and designed the experiments, performed the experiments, analyzed the data, prepared figures and/or tables, authored or reviewed drafts of the paper, and approved the final draft.
- Jesús Genaro Sánchez-Martínez and Jaime Luis Rábago-Castro performed the experiments, analyzed the data, authored or reviewed drafts of the paper, and approved the final draft.
- Flaviano Benavides-González, María de la Luz Vázquez-Sauceda and Lorena Garrido-Olvera analyzed the data, authored or reviewed drafts of the paper, and approved the final draft.

### Field Study Permissions

The following information was supplied relating to field study approvals (i.e., approving body and any reference numbers):

Sampling in waters under federal jurisdiction was approved by CONAPESCA through permit number: DGOPA/05675/060505/-3869.

## Data Availability

The raw data including biotic and abiotic variables registered within two seagrass beds in a coastal lagoon, shrimp size and abundance (*Penaeus aztecus* and *P. duorarum*), and estimates of shrimp condition by seagrass bed are available in Supplementary Files.

## Supplemental Information

Supplemental information for this article can be found online at http://dx.doi.org/10.7717/peerj.10496#supplemental-information.

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
