# Peer review of "Density-dependent condition of juvenile penaeid shrimps in seagrass-dominated aquatic vegetation beds located at different distance from a tidal inlet"

_PeerJ, doi:10.7717/peerj.10496_

## Round 0.1 · original submission · Major Revisions

Your paper has to be deeply improved. Some important information is missing (see comments of reviewers).

Reviewer 1 ·

Basic reporting

No comment

Experimental design

No comment

Validity of the findings

No comment

Additional comments

This is a well-written study in which the authors report density-dependent effects in two species of penaeid shrimp inhabiting seagrass beds located at different distances from a tidal inlet in Laguna Madre, Mexico. This was a field study, shrimp samples were taken in triplicate through three tows in each seagrass bed. Salinity, temperature and dissolved oxygen were recorded in each seagrass bed. The sampling method seems correct, as well as analytical and statistical methods.Their results show a negative relationship between body condition and density, with differences related to the proximity of the tidal inlet.
The authors discussed that the proximity of the tidal inlet is a crucial factor in shrimp density, in which the nearby seagrass bed showed higher density and poorer body condition than the distant seagrass bed.
While the proximity of the tidal inlet could indeed be a factor, the authors also report significant differences in salinity and macroalgal biomass between the two locations, but no analysis was performed (or at least it is not shown) to see if any of these factors also play a significant role in density and condition. I would suggest performing this analysis with the data presented in supplementary files 1, 3, and 5 just to know if other factors (besides proximity to the tidal inlet) participate in the density-dependent effect observed in this study.

Reviewer 2 ·

Basic reporting

The Introduction, Methods and Results are well presented. The grammar is clear are well structured. Some sentences would be improved with a minor restructure if the grammar. I have annotated a few examples in the PDF of the paper.

The grammar in the Discussion is less well structured. Portions of the discussion drift and seem to suffer from a lack of clear points to reference. The authors could define two to three critical themes and construct the discussion around them.

The first paragraph (lines 220-224) discusses shrimp condition, then paragraph 5 (lines 254-270) discusses shrimp condition again, and relate the current study to past studies. These two paragraphs could be consecutive. The reasons for different shrimp density (differing postlarval advection and settlement) and the intraspecific and interspecific competition (food scarcity) can then be discussed. But see comments on settlement, recruitment and shrimp size class elsewhere in this review.

Paragraph 4 (lines 245-253) is unconvincing.

The Conclusions are generally good, however, the statement that proximity to the inlet positively influenced the abundance of shrimp is not able to be supported by these data. Proximity is related to shrimp abundance only.

The References comply with Peer J format.

Experimental design

The experimental design is clear.

A criticism of the design is that the data series was conducted for one year only; two or three years would have been more appropriate.
The most robust data gathered by this work are the extensive length/weight data sets for each species caught in each seagrass bed; the ‘distant’ and ‘nearby’ seagrass beds. These data, together with the density data for each seagrass bed, are enough to support the investigation and conclusions of this paper. These data should be the bedrock of the paper.

The month-to-month, size step-up analyses undertaken are the least robust data presented. These data would benefit from a longer sampling period. Two years of data would provide 23 data points; three years would provide 35 data points. Each case would provide a more robust correlation series than the 11 data points currently analysed.

A description of the tide range and the likely tidal currents in Laguna Madre is needed (or data of available). This would assist understanding the capacity for within-lagoon mobility by postlarvae using tidal currents. For example, if tide range was small and if tidal exchange and current strength was low, then the case put by the authors would have more support.

As well, a description of the extent and possibility of microhabitats of the seagrass community in Laguna Madre would benefit the paper. Has habitat mapping in general and the seagrass extent within Laguna Madre been undertaken? Do seagrasses occupy different depth strata? Have previous studies demonstrated that the current seagrass habitats sampled are the key settlement habitats for shrimp postlarvae?

Sampling Gear
The study techniques (net gear) do not capture the postlarval life-history stage of the shrimp as they would escape the 13 mm mesh. Loneragan et al. (1994) and Vance et al. (1990) sampled postlatval and juvenile shrimp in seagrass and mangrove habitats and used a 2 mm mesh net (1 mm mesh cod end) in which prawns from 1 mm CL to about 12 mm CL were caught efficiently. Hence, if the authors were attempting to sample ‘recruitment’ at each of the ‘distant’ and ‘nearby’ seagrass habitats they don’t achieve their aim. Currently their net configuration allows the 1 to ~5 mm CL animals to escape the net through the mesh. The unsuitability for the 13 mm mesh nets to catch recently settled postlarvae and very small juvenile shrimp is clear. In escapement of small shrimp from the nets has been documented by Pérez-Castañeda and Defeo (2005) who state that <30% of shrimp < 8 mm CL are retained by this type of net.

Note that the size structure of shrimp populations in shallow coastal seagrass communities is highly skewed to the smallest size classes. In Australia, coastal populations of a seagrass-dwelling shrimp (P. semisulcatus) demonstrate that 60-70% of the population is in the 1-3 mm CL range (80-90% is in the 1-6 mm CL range) (Vance et al. 1996, Figure 7; Loneragan et al. 1994, Figure 6), yet this size range is not efficiently sampled with the gear used during the current study. These 1-2 mm CL shrimp are the critical first recruits to inshore seagrass habitats.

Clearly, shrimp in the 1-6 mm CL size range could have settled in high abundances in each of these seagrass habitats and post-settlement processes (predation, competition, migration, mortality) could have reduced numbers differentially so that by approximately 6-8 weeks after settlement whereby the shrimp reached 7-10 mm CL, their abundance had nothing to do with recruitment, but other processes. Perhaps there is strong and equivalent recruitment at each seagrass location, yet a collapse of the 2-5 mm CL size range of shrimp at one site and not the other. The 13 mm mesh nets would not detect these possible aspects of a skewed population structure.
In addition, penaeid shrimp postlarvae can use microhabitats that are closeby to habitats used by larger size classes, but they do not inhabit the exact same spatial distribution, and it may take years to identify the non-overlap (Vance et al. 1990). Within the same seagrass bed, the shallowest portions of the habitat (0.5 m deep at low tide) can be the critical settlement habitat of postlarvae where they are exclusively abundant. While medium-sized juveniles are found in both the shallower and deeper seagrass extents (Loneragan et al. 1994). It is wise to sample a range of depth strata in seagrass habitats to ensure all shrimp size ranges are sampled and critical habitats that are limited in extent and scope are identified and sampled. Apart from seagrass species and aboveground abundance, only depth of the two seagrass habitats in a range of 1.0 to 1.5 m is mentioned as to the fine scale description of the seagrass habitat. It is possible that, not only were postlarvae escaping the nets, the critical depth strata of postlarval settlement and recruitment may not have been sampled.

As a consequence, this paper does not clearly show that distance from the mouth of this lagoon is critical to recruitment difference between the two seagrass sites. Distance is related to the difference, but not necessarily determinant. Nets set in the water column to catch pelagic larvae entering the lagoon and moving ‘up-lagoon’ in the vicinity of each site (1 km and 25km) would elucidate quanta of potential recruitment at each site. The sampling of 1-2 mm CL recent recruits also would provide critical information.

This facet highlights a point made clearly in this review. The length /bodyweight weight relationship is the key dataset that should underpin this paper. The authors have 1000’s of measured and weighed shrimp; samples from each habitat and a difference in condition between the two habitats separated by 25 km. Robust determinations and conclusions can be made from these data, and should be. Relating month-to-month size categories to growth and cohort progression at different shrimp densities from these data is the least conclusive aspect of this study.

Raw data
The most robust datasets described in this paper are the ~2000 and ~ 1000 length/weight data for the two species of penaeid shrimp. There are 685 individual P. aztecus from the distant seagrass bed and 1426 individual P. duorarum from the nearby seagrass bed. Similarly, there were 258 individual P. aztecus from the distant seagrass bed and 686 individual P. duorarum from the nearby seagrass bed. Yet these data are not emphasized in the results or the discussion. The stated aim of this paper could be achieved using these data alone. Other analyses are subsequent to the establishment of a difference in shrimp condition.

However, this analysis is mentioned at line 155 of the document. The exploration of mean abundance of ‘small’ and ‘large’ shrimp by month, and their supposed contribution of the abundance of ‘small’ shrimp to the abundance of ‘large’ shrimp the next month is mentioned in line 135 of the text.

The order of mention of these analyses should be reversed. The paper depends of a difference in length/weight relationships within the ‘nearby’ and ‘distant’ seagrass beds as a precursor to subsequent analyses.

Several analyses are described in lines 162 to 173; improved delineation and explanation of the logic behind these (at least) three analyses is required.

Conduct cohort analyses such as O’Brien (1994) in each of the ‘distant’ and ‘nearby’ seagrass populations to investigate the growth rates in each habitat type.

Validity of the findings

Similar to the perspective from the methods, the bedrock dataset for this paper is the length/weight data form the two species of shrimp. Previously, length weight relationships have been used to asses prawn ‘condition’ seasonally and in different habitats (Pérez-Castaňeda, R. and Defeo, O, 2002, Ochwada-Doyle et al. 2011). The presentation of data by Ochwada-Doyle clearly demonstrates the primacy of the length/weight data to establish the relationships that underpin assessment of condition in distinct habitats. Their Figure 1 clearly shows the information content of the data to demonstrate Ochwada-Doyle et al. (2011) used the accumulation of each individual shrimp’s length and weight to describe and compare relationships for shrimp condition between habitats. That is, linear correlations were determined for shrimp caught in each habitat type and hundreds of animals were included in each correlation. The large number of data points provided a robust relationship between length and weight and the slopes of the linear relationship could be compared (R > 0.9 in all cases).

Currently, the authors of this paper (Blanco-Martínez et al.) focus attention on their month to month density estimates; presuming a cohort of ‘large’ shrimp represents a cohort of ‘small’ individuals that have grown in size from the previous month. The annual sampling provides 11 data points. The analyses rely on the 11 data-points from which to determine a relationship. For both the ‘distant’ and ‘nearby’ habitats, the scatter plots of the data show a cluster of points in the lower left of the plot, and a scatter of points over the rest of the plot. These data represent a single year’s sampling. The analyses determine significant relationships among the data, however, the R2 for each correlation is not strong, both <0.5. The analyses would benefit from another year‘s data (or two years (~30 data points)) to put rigour into the relationship.

The focus of the Results should be the ~2000 and ~ 1000 length/weight data for the two species of penaeid shrimp. There are 685 individual P. aztecus from the distant seagrass bed and 1426 individual P. duorarum from the nearby seagrass bed. Similarly, there were 258 individual P. aztecus from the distant seagrass bed and 686 individual P. duorarum from the nearby seagrass bed. The stated aim of this paper could be achieved using these data alone. Other analyses are subsequent to the establishment of a difference in shrimp condition.

The research reported in the paper fails to measure true recruitment to the ‘distant’ and ‘nearby’ seagrass habitats in Laguna Madre. There is an extensive literature on settlement and recruitment; and the exact definition of each process remains discussed. In part, it depends on the taxa being described. Generally for penaeid shrimp, settlement refers to the morph from a pelagic lifeform to a benthic lifeform; while recruitment can refer to the ‘success’ of recently-settled individuals surviving an initial time-period as benthic individuals. Either way, in the case of penaeid shrimp postlarvae/early juveniles, recruitment involves animals in the 1 to 3 mm CL size range. These are true recruits (even 1-2 mm CL individuals) that are new residents in a seagrass habitat and soon afterwards will be subject to predation, competition, migration etc. within the seagrass.

Juvenile P. aztecus and P. duorarum will be subject to ~10-20% mortality wk-1 (Minello et al. 1989, O’Brien 1994, Wang and Haywood 1999) in seagrass habitats prior to their reaching the 7-15 mm CL size range that is the focus of this paper. They recruit at 1-2 mm CL, survive predation and competition for > 4-6 weeks and may move microhabitat before they are sampled as part of this study.

The authors need to rewrite the Discussion acknowledging these issues.

Figures

Figure 5 is nonsensical. It plots predicted mean weight against total abundance for the ‘distant’ and ‘nearby’ seagrass habitats grouped, and suggests a relationship is evident. The basic premise of the study is that there is no weight/abundance relationship in the ‘distant’ seagrass bed, yet there is a relationship in the ‘nearby’ seagrass bed; it is nonsensical to use these data in the same plot and explore a relationship.

Additional comments

This study provides robust data to assess the condition of two species of shrimp in two seagrass habitats in a relatively narrow, elongated lagoon with a single narrow opening to the ocean. Condition is defined as the relationship between shrimp length (mm CL) and body weight, where a heavier shrimp for a given length is in better ‘condition’. One seagrass bed is closeby to the inlet, one is 25 km distant; they are similar in the vegetated found at each site, the authors identify distance from oceanic source of shrimp postlarvae as the key factor determining different characteristics of the shrimp population between the two sites.

Lines 82-91 of the introduction make distance from the ocean inlet, the source of pelagic postlarvae, as a key determinant of ‘recruitment’ and hence shrimp abundance. Higher or lower abundance then is postulated as the driver of shrimp condition.

However, the study does not sample shrimp recruitment. The key size classes of postlarval recruits (1-2 mm CL) escape from the relatively large meshed nets (13 mm) used during the study. Dall et al. (1990) include P. aztecus and P. duorarum among their Type 2 or 3 penaeid crustaceans with an offshore spawning, pelagic larval and postlarval phase that advect inshore to settle to a benthic existence in critical shallow coastal nursery habitats, and subsequent ontogenetic emigration back offshore. The current study highlights 7 to 10 mm CL and 12-15 mm CL shrimp as the focus size ranges analysed. Shrimp less than about 7 mm CL are poorly represented among the size classes of the catches. Hence the critical ‘settlers and recruits’ are missed by the gear used to sample the shrimp.

Shrimp in the 7-15 mm CL size range are residents of seagrass habitats, not recruits. 15 mm CL animals are upcoming candidates for ontogenetic emigration to adult habitats. These are large juveniles that have been resident on the seagrasses for at least 6-8 weeks, and subject to predation, migration, mortality etc. Zimmerman and Minello (1984) and Minello et al. (1989) studied the population dynamics of juvenile P. aztecus in the Gulf of Mexico in the 1980s and showed that shrimp from 10-30 MM total length (TL) (~1 to 4 mm CL) form the bulk of the population on a seagrass bed. The basic biology of these species demonstrates that these sizes would be present and the critical phase for recruitment in Laguna Madre. For a similar seagrass-dependent penaeid shrimp on the other side of the world, O’Brien (1994) provides a detailed breakdown of the size range of juvenile P. esculentus on a seagrass bed (Figure 6), including modal growth analysis of a series of cohorts during the recruitment season of the species. The dominance of the smallest size classes of shrimp is clear (1-4 mm CL) and the quick demise of each cohort as the individuals grow and reach the size classes that are the focus of this paper (8-15 mm CL) is repeated for subsequent cohorts through the spring/summer seasons.

Despite these criticisms, the current study provides valid data on prawn condition in two seagrass habitats. It indicates that distance from the oceanic source of the earliest phase of their life history is related to condition and abundance (density) of shrimp within each habitat. The current study cannot attribute causality to this relationship and does not measure differential recruitment between the seagrass habitats. The authors need to reconsider the themes underpinning their paper and be clear about what their data demonstrate.

There is a history of juvenile shrimp population studies in coastal lagoon habitats of the Gulf of Mexico that use nets with a relatively large mesh size to sample juvenile shrimp (Pérez-Castañeda and Defeo 2002, 2005). This gear may be appropriate for the aims of many studies, but for the current study that concludes that recruitment is the key driver of density and hence shrimp condition, the gear is not appropriate as it does not sample the recruits. It samples mid-aged resident juveniles, the population of which is the result of many post-settlement processes. The authors need to make this point clear, then continue with their description of condition in two seagrass habitats and then it is valid to hypothesize that distance from the source inlet may contribute to variation in density and hence condition.

Rather than emphasizing the supply of postlarvae alone as the source of variation in abundance and condition between the distant and nearby habitats, the shrimp size classes represented here could couched as representative of the interrelation of growth, mortality and migration and this ‘successful recruitment’ to these habitats of individuals who have survived and likely will contribute to the offshore adult population. The second main analysis employed in this study attempts to validate this type of theme. It relates subsequent monthly cohorts of two shrimp size classes as a statistically valid progression. That is, shrimp in the 7-10 mm CL size class were supposed to grow to form the 12-15 mm CL size class in the subsequent month. The monthly data plotted to represent this relationship are correlated. While there is logic to this relationship, only 11 data points are analysed (1 year’s data) and the correlation R2 is < 0.50. This analysis would have benefitted from a two or three year data series to provide a more robust dataset to correlate. These analyses are secondary to the length/weight relationships.

The discussion of the paper requires at least two distinct themes to build a narrative around. The authors need to justify why they do not sample the ‘true-recruits’ within these habitats. Reference to the early studies of Minello and Zimmerman (see Minello et al. 1989, Figure 1) and their colleagues provide explicit detail on the structure and dynamics of one of the species studied here, including quantification of the full range of size classes found in inshore habitats (see Figure 1), yet the mention of this historical cache of literature is scant. Studies by Haywood and Staples (1993), O’Brien (1994), Haywood et al. (1998) and Wang and Haywood (1999), provide descriptions of robust analyses of growth and mortality of seagrass-dependent penaeid species (and others), yet only one paper is mentioned briefly. Best to address the issue of what is recruitment and build an argument from that. Ochwada-Doyle et al. (2011) provide a clear example of the use of condition assessment in determining the ‘health’ of prawns that are released as part of artificial stock enhancement. Yet this is a theme that is not explored. These animals are medium-sized juveniles as well, similar to the size classes investigated in this paper.

This paper has merit; yet the authors need to build a basis for their concept of recruitment, back-up their focus on the size-class-limited medium-size juvenile shrimp and crystalise an argument around these themes and the usefulness of the outcome of their study.

Literature referred to:
Dall, W., Hill, B. J. Rothlisberg, P.C. and Staples, D. J. (1990). The biology of the Penaeidae. Advances in Marine Biology 27.

Haywood, M. D .E., and Staples, D. J. (1993). Field estimates of growth and mortality of juvenile banana prawns (Penaeus merguiensis). Marine Biology 116, 407.16.

Haywood, M. D .E., Heales, D. S., Kenyon, R. A., Loneragan, N. R. and Vance, D. J. (1998). Predation of juvenile tiger prawns in a tropical Australian Estuary. Marine Ecology Progress Series 162, 201-214.

Loneragan, N. R., Kenyon, R. A., Haywood, M. D .E. and Staples, D. J. (1994). Population dynamics of juvenile tiger prawns (Penaeus esculentus and P. semisulcatus) in seagrass habitats of the western Gulf of Carpentaria. Marine Biology 119, 133-143.

Minello, T. J., Zimmerman, D. J. and Martinez, E.X. (1989). Mortality of young Brown Shrimp (Penaeus aztecus) in estuaries nurseries. Transactions of the American Fisheries Society 118, 693-708.

O’Brien, C. J. (1994). Population dynamics of juvenile tiger prawns Penaeus esculentus in south Queensland, Australia. Marine Ecology Progress Series 104, 247.56.

Ochwada-Doyle, F., Gray, C.A., Loneragan, N.R., Taylor, M. D. and Suthers, I.M. (2011) Spatial and temporal variability in the condition of postlarval and juvenile Penaeus plebejus sampled from a population subject to pilot releases. Aquaculture Environment Interactions 2, 15-25.

Pérez-Castañeda, R. and Defeo, O., 2002. Morphometric relationships of penaeid shrimpin a coastal lagoon: spatio-temporal variability and management implications. Estuaries, 25(2), 282–287.

Pérez-Castañeda, R. and Defeo, O., 2005. Growth and mortality of transient shrimp populations (Farfantepenaeus spp.) in a coastal lagoon of Mexico: role of the environment and density-dependence. ICES Journal of Marine Science, 62(1), 14–24.

Vance, D.J. Haywood, M. D .E. and Staples, D. J. (1990). Use of a mangrove estuary as a nursery area by postlaval and juvenile banana prawns (Penaeus merguiensis) de Man, in northern Australia. Estuarine, Coastal and Shelf Science 31, 689-701.

Vance, D.J., Haywood, M. D .E., Heales, D. S. and Staples, D. J. (1998). Seasonal and annual variation in abundance of postlaval and juvenile tiger prawns Penaeus semisulcatus and environmental variation in the Embley River, Australian: a six year study. Marine Ecology Progress Series 135, 43-55.

Wang, Y.-G., and Haywood, M. D. E. 1999. Size-dependent natural mortality of juvenile banana prawns Penaeus merguiensis in the Gulf of Carpentaria, Australia. Marine and Freshwater Research, 50: 313-317.

Zimmerman, D. J. and Minello, T. J (1984). Densities of Penaeus aztecus), , and other natant macrofauna in a Texas salt Marsh. Estuaries 7, 421-433.

Annotated reviews are not available for download in order to protect the identity of reviewers who chose to remain anonymous.

---

## Round 0.2 · accepted · Accept

Thank you very much for improving your manuscript and for submitting your work to this journal.

Reviewer 1 ·

Basic reporting

In this revised version of the manuscript the authors made major modifications and it improved substantially. The English language is correct and professional, and references are appropriate and sufficient for the contents of the study.

Experimental design

The experimental design is well explained, with sufficient detail, as well as methods and statistical analysis.

Validity of the findings

Findings are supported by the data, the discussion is well presented and adequate according to the findings, it is not speculative, conclusions are well stated.

Additional comments

This revised version has improved compared to the previous one, the authors have answered my questions and I am satisfied with the answers. The manuscript is well written although I spotted two minor mistakes (line 335, “Shrimp abundance peaked at the beginning…” (instead of beginnin), and line 540, “However, it would be necessary…” (instead of It). Other than that I believe the article is ready for publication.

Reviewer 2 ·

Basic reporting

The Introduction, Methods and Results are well presented. The grammar is clear are well structured. A few suggestions have been made as tracked changes in a Word version where previous tracked changes have been accepted.

The grammar in the Discussion is much improved and mostly is well structured.

The Discussion should be edited to improve its grammatical structure and composition (minor issues). A few suggestions have been made as tracked changes in a Word version where previous tracked changes have been accepted.

The References comply with PeerJ format.

Experimental design

The experimental design is clear.
A criticism of the design is that the data series was conducted for one year only; two or three years would have been more appropriate.

The most robust data gathered by this work are the extensive length/weight data sets for each species caught in each seagrass bed; the ‘distant’ and ‘nearby’ seagrass beds. These data, together with the density data for each seagrass bed, are enough to support the investigation and conclusions of this paper. These data have been analysed and discussed as the bedrock of the paper.

A description of the tide range and the likely tidal currents in Laguna Madre has been provided. This information assists to understand the capacity for within-lagoon mobility by postlarvae using tidal currents.

At the project conception phase, a strategy that successfully sampled recruitment of pelagic postlarvae to their benthic stage and their short-term success colonising each of the closeby and distant seagrass habitats would have benefitted the paper.

Validity of the findings

The focus of this paper’s Results has been shifted to be the ~2000 and ~ 1000 length/weight data for the two species of penaeid shrimp. There are 685 individual P. aztecus from the distant seagrass bed and 1426 individual P. duorarum from the nearby seagrass bed. Similarly, there were 258 individual P. aztecus from the distant seagrass bed and 686 individual P. duorarum from the nearby seagrass bed. The stated aim of this paper has been achieved using these data alone.

The critical data for this paper are the length/weight data form the two species of shrimp. Previously, length weight relationships have been used to asses prawn ‘condition’ seasonally and in different habitats (Pérez-Castaňeda, R. and Defeo, O, 2002, Ochwada-Doyle et al. 2011). The presentation of data by Ochwada-Doyle clearly demonstrates the primacy of the length/weight data to establish the relationships that underpin assessment of condition in distinct habitats.

The Discussion is much improved and reads well. It provides a good examination of the results of the current study relative to past work. It also suggests that the outcomes of this research assist an understanding of the contribution of different spatial locations of ‘occupied juvenile shrimp habitat’ within Laguna Madre to the oceanic populations of P. duorarum and P. aztecus. This relationship allows further studies to be conceptualized.

The Discussions also highlights the gap-in-knowledge about actual postlarval settlement and recruitment within Laguna Madre and how both tidal circulation/movement and seagrass habitat structural complexity may influence the resultant population of juvenile shrimp in these lagoon seagrass habitats, and hence their condition, survival and emigration offshore. This knowledge gap allows further studies to be conceptualized.

Additional comments

The authors have responded to review by addressing the issues raised. They have removed the contention that ‘recruitment’ was measured in each seagrass habitat separated by distance from the ocean inlet and its presumed source of pelagic postlarvae. In addition, they have removed the month-to-month cohort progression and the statistical justification of that size-progression link.

In its current form, this paper has merit and meets a standard of scientific rigour for the experiment undertaken. The authors have focused on the size-class-limited medium-size juvenile shrimp and presented an argument for the usefulness of the outcome of the assessment of shrimp condition from their study. The body of work described in the paper would have benefitted from additional sampling of smaller stages in the life history of penaeid shrimp to conclude a more comprehensive and cutting-edge study. At the conceptual stage and determining the hypotheses to be tested, the limitation of field gear that efficiently caught shrimp > ~ 7 mm CL should have become evident. The limitation being that first-recruits of penaeid shrimp (~2-3 mm CL) would not be retained in the net by 13 mm mesh gear. The experiments described in the paper would have benefitted from a sampling regime that did sample recruitment at each seagrass habitat, as well as the condition of shrimp encountered.

The authors could have undertaken pelagic plankton tows in the vicinity of each of the seagrass habitats in part to test their hypothesis that the potential for ‘recruitment’ was significantly greater closeby to the ocean inlet. They could have used a small mesh net (~2mm mesh) in conjunction with the 13 mm mesh net that they used to sample the postlarvae and smallest juvenile phases of the shrimp on the seagrasses. Field data that estimated actual recruitment of 2-3 mm CL shrimp or potential recruitment (1-2 mm CL Postlarvae) would have complemented the data presented in the paper and ensured much more interest in the author’s body of work from shrimp biologists worldwide.

Annotated reviews are not available for download in order to protect the identity of reviewers who chose to remain anonymous.